# Renal cancer cells acquire immune surface protein through trogocytosis and horizontal gene transfer

Haley Q. Marcarian[1,2], Anutr Sivakoses[1,2], Anika M. Arias[1], Olivia C. Ihedioha[2], Benjamin R. Lee[3], Maria C. Bishop[4], Alfred L. M. Bothwell[1,2,5]*

1 Cancer Biology Graduate Program, University of Arizona, Tucson, Arizona, United States of America, 2 Department of Pathology, Microbiology, and Immunology, University of Nebraska Medical Center, Omaha, Nebraska, United States of America, 3 Department of Urology, University of Arizona, Tucson, Arizona, United States of America, 4 College of Medicine, University of Arizona, Tucson, Arizona, United States of America, 5 Department of Immunobiology, Yale University School of Medicine, New Haven, Connecticut, United States of America

* albothwell@unmc.edu

## Abstract

Trogocytosis is an underappreciated phenomenon that shapes the immune microenvironment surrounding many types of solid tumors. The consequences of membrane-bound proteins being deposited from a donor immune cell to a recipient cancer cell via trogocytosis are still unclear. Here, we report that human clear cell renal carcinoma tumors stably express the lymphoid markers CD45, CD56, CD14, and CD16. Flow cytometry performed on fresh kidney tumors revealed consistent CD45 expression on tumor cells, as well as varying levels of the other markers mentioned previously. These results were consistent with our immunofluorescent analysis, which also revealed colocalization of lymphoid markers with carbonic anhydrase 9, a standard kidney tumor marker. RNA analysis showed a significant upregulation of genes typically associated with immune cells by tumor cells. Finally, we show evidence of chromosomal DNA being transferred from immune cells to tumor cells through physical contact. This horizontal gene transfer has transcriptional consequences in the recipient tumor cell, resulting in a fusion phenotype that expresses both immune and cancer specific proteins. This work demonstrates a novel mechanism by which tumor cell protein expression is altered through the acquisition of surface membrane fragments and genomic DNA from infiltrating lymphocytes. These results alter the way in which we understand tumor-immune cell interactions and may reveal new insights into the mechanisms by which tumors develop. Additionally, further studies into trogocytosis and other mechanisms of contact-mediated cellular transfer will help push the field towards the next generation of immunotherapies and biomarkers for treating renal cell carcinoma and other cancers.

**Data availability statement:** All relevant data are within the manuscript and its Supporting information files.

**Funding:** We acknowledge use of the University of Nebraska Medical Center - UNMC Advanced Microscopy Core Facility, RRID:SCR_022467, P20 GM103427 (NIGMS, NE-INEBRE), P30 GM106397 (NIGMS, NCS), P20GM130447 (NIGMS, CoNDA), P30 CA036727 (NCI, Buffett Cancer Center), S10RR02730 (NIH), S10OD030486 (NIH), Nebraska Research Initiative, UNMC Vice Chancellor for Research Office. Research reported in this publication was supported by the National Cancer Institute of the National Institutes of Health under award number P30 CA023074. The UNMC Flow Cytometry Research Facility is administrated through the Office of the Vice Chancellor for Research and supported by state funds from the Nebraska Research Initiative (NRI) and The Fred and Pamela Buffett Cancer Center's National Cancer Institute Cancer Support Grant (P30 CA036727).

**Competing interests:** No competing interests to declare.

## Introduction

Trogocytosis is the transfer of membrane fragments from a donor cell to an acceptor cell [1]. In mammalian cells, this process was first described on CD8[+] T cells, in which the CD8[+] T cells were shown to take fragments of the plasma membrane from certain antigen presenting cells (APCs) during the formation of the immunological synapse [2]. This can result in activated T cells acquiring MHC class I molecules loaded with peptide (pMHC class I) and subsequent fratricide of pMHC class I-presenting T cells by other activated T cells in culture [3]. Since then, trogocytosis has been observed in many different types of immune cells, including CD4[+] T cells [4], Natural Killer (NK) cells [5], marginal zone B cells [6], and macrophages [7]. Previously, we have shown that trogocytosis plays an important role in the acquisition of immune regulatory molecules by colon cancer cells from infiltrating lymphocytes [8]. This resulted in increased immune regulator molecules such as CTLA4, PDL1, Tim3, VISTA, LAG3, CD38, CD80, CD86, MHC Class II, and PD-L1 being presented on the surface of trogocytic tumor cells compared to non-trogocytic tumor cells. Trogocytosis was also observed with numerous lymphocyte populations including macrophages, dendritic cells, NK cells, and monocytes [8].

Trogocytosis has also been implicated in other types of disease such as Hodgkin lymphoma and renal cell carcinoma [9,10]. In mouse models of leukemia, trogocytosis promotes the transfer of target antigens from cancer cells to T cells, which induces T cell exhaustion and fratricide [11]. T cell exhaustion and fratricide often cause relapses in patients treated with CAR-T cell therapies. Recently, researchers have shown that fusing CTLA-4 cytoplasmic tails to the c-terminus of the chimeric antigen receptor on CAR-T cells reduces trogocytosis induced fratricide by promoting CAR recycling [12]. However, the extent to which trogocytosis occurs in tumors as well as the implications it has in shaping the tumor-immune microenvironment have not been fully elucidated.

Horizontal gene transfer (HGT) is a process that has been widely studied in the field of microbiology. HGT between different species of bacteria has been shown to grant survival advantages to the recipient microbe, such as resistance to antibiotics [13]. However, the concept of horizontal gene transfer between mammalian cells is largely uninvestigated. Recently, it has been shown that RPE1 cells, an immortalized non-transformed pigmented epithelium line can integrate a genomically encoded fluorescent reporter from a breast cancer cell line expressing that reporter when the two lines are cocultured [14]. Interestingly, physical contact was required between the two cell lines for HGT to occur. The transfer of intracellular contents, including genomic DNA, being a mechanism for cancer development and progression has garnered much interest in recent years, with some regarding it as a novel hallmark of cancer [15]. This transfer may occur through several mechanisms, including cell fusion, entosis and cell cannibalism [16]. Researchers have shown that spontaneous cell fusions between two different cancer cell lines can result in parasexual diversification in tumor cell populations, potentially leading to more aggressive disease [17]. However, this phenomenon has not been well studied in human models of disease, particularly concerning cell fusion events between cancer cells and tumor infiltrating lymphocytes.

In this study, we detected the presence of the immune markers CD14, CD16, CD56, and CD45 on tumor cells from renal cell carcinoma (RCC) patients. Previously, these markers have not been identified on RCC cells or other solid tumors. We suggest they were acquired through trogocytosis of infiltrating immune cells as these tumors develop. Other markers, including CD19 and CD11b were also tested and determined to not be significant markers of trogocytosis. These results were determined through quantification of immunofluorescent staining and flow cytometry. RNA analysis further showed contact between RCC cells and infiltrating lymphocytes resulted in an altered phenotype of tumor cells that expressed both tumor-specific and immune-specific genes. NanoString analysis of trogocytic tumor cells from human RCC tumors showed that trogocytic cells display rampant expression of immune cell-associated and tumor-associated genes. Finally, *in vitro* analysis of EdU and a genomically encoded GFP tag transfer from T cells to RCC cells showed that genomic DNA is transferred when these cells are in contact. Importantly, the effects of this transfer are ablated when the cells are separated by a transwell barrier.

## Materials and methods

### Immunofluorescent staining assays

The following primary antibodies were used in immunofluorescent experiments: anti-human CAIX (R&D Biosystems, AF2188), CD45RA (Sino Biological Inc., 102580-T08), CD56 (ProteinTech®, 14255-1-AP), CD68 (Sino Biological Inc., 11192-T26), CD16 (ProteinTech®, 16559-1-AP), CD14 (R&D Biosystems, BAF383). Paraffin-embedded tissue slides were obtained from Tissue Acquisition and Cellular/Molecular Analysis (TACMASR) core facility at the University of Arizona. Slides were deparaffinized for 9 min in Xylenes. They were then washed in decreasing concentrations of ethanol solutions at 1 min intervals for 5 min. The slides were then washed in MilliQH$_2$O for 10 minutes. Then, antigen retrieval was performed using boiling Antigen Unmasking Solution (Vector Laboratories, H-3300) for 10 min and left to cool for an additional 20 min. Slides were blocked with 20% donkey serum/1X DPBS-0.1% Tween-20 for 30 min. Primary antibody solutions were prepared according to their recommended dilutions. Slides were incubated with primary antibodies overnight at 4°C and washed with 1X DPBS-0.1% Tween-20 for 5 min. Samples were quenched using Invitrogen ReadyProbes™ Tissue Autofluorescence Quenching Kit for 3−5 min. Secondary anti-donkey antibodies from Invitrogen were applied and incubated for 1 hour at room temperature in the dark. Samples were quenched an additional time and stained with Hoescht nuclear stain for 10 min. Coverslips were mounted on the stained slides using Fluoromount-G™ (Invitrogen, 00-4958-02). Slides were imaged using the Echo Inc. Revolution fluorescent microscope.

### Flow cytometry assays

The following conjugated antibodies were used in flow cytometric experiments: CD45 (clone 2D1, BioLegend, 368532), CD16 (clone CB16, Invitrogen, 17-0168-42), CD14 (clone 61D3, Invitrogen, 414-0149-42), CD45RO (clone UCHL1, Invitrogen, 12-0457-42), CD45RA (clone HI100, Invitrogen, 17-0458-42), CD44 (clone IM7, Invitrogen, 404-0441-82), CAIX (R&D Biosystems, FAB2188G), CD56 (clone CMSSB, Invitrogen, 12-0567-42), and CD68 (clone REA886, Miltenyi Biotec, 130-114-463). Fresh tumor tissue was acquired through the Tissue Acquisition and Cellular/Molecular Analysis (TACMASR) core facility at the University of Arizona. Tumor samples were finely minced into 1mm$^2$ sections and washed with DPBS. They were then suspended in a solution of 1 mg/ml Collagenase Type 4 (Worthington, LS0004186) and incubated at 37°C for 30 min, being shaken every 3 min. The reaction was stopped using DMEM media with 5% fetal bovine serum. Cell suspensions were then filtered through a 40μm mesh filter and resuspended in 100μl of DPBS. Cell viability was stained with Ghost Dye™ Violet 450 (Tonbo Biosciences, 13-0863-T100). Human BD Fc Block™ (BD Pharmingen™, 564220) was used at a concentration of 5μl/10$^6$ cells for 10 min at room temperature. Fluorescent antibody staining was done at the recommended concentrations of each antibody for 30 min at room temperature in the dark. Cell suspensions were then washed using DPBS and resuspended in 1X RBC lyse/fix solution (eBioscience™ 00-5333-57) for 40 min. Samples were then washed in DPBS and analyzed using the BD FACSCanto™ II system. Data was analyzed using FlowJo v10.8.1.

## NanoString preparation and analysis

Fresh human ccRCC tissue samples were prepared for flow cytometry and labeled with anti-CD45 and anti-CAIX antibodies (CD45 (clone 2D1, BioLegend, 368532), CAIX (R&D Biosystems, FAB2188G)). Samples were sorted using a BD FACSAria II cell sorting system. RNA was extracted from sorted samples using the RNeasy Micro Kit (Qiagen, 74004) according to manufacturer's guidelines. Quality control and preparation for NanoString analysis was performed by the Arizona Genetics Core facility at the University of Arizona. Data analysis was done using nSolver Analysis Software 4.0. Background thresholding was performed using the geometric mean value of negative control probes. CodeSet Content Normalization was calculated using the geometric mean count of housekeeping genes. Pairwise comparisons were calculated between sample groups.

## Immunofluorescent image quantification

Images used for analysis in Fig 1 were taken at 40X (NA 1.40) on an Echo Revolution LED microscope. Images were exported in separate channels into ImageJ (FIJI) for quantification. Nuclei were identified using the StarDist2D plugin [18], then a Voronoi algorithm was applied to estimate the boundaries of individual cells. The tumor marker channel was used to subtract all non-tumor cells from the Voronoi cell map. The resulting regions of interest (ROIs) were then saved to be used as a map for all tumor cells in each image. The channel containing the trogocytic marker of interest was then run through a background subtraction algorithm and the resulting signal was normalized using the CLAHE histogram normalization plugin. The channel was then converted to a binary image and the ROIs previously described were applied. The total surface area of each cell was measured to give a two-dimensional representation of the cellular surface area being occupied by a trogocytic marker. Cells with trogocytic markers occupying over 55% and less than 90% of their total 2D surface area were considered trogocytic. These cutoffs were determined to exclude autofluorescence, other sources of background signal, and infiltrating immune cells from quantification. We believe this method of quantification is quite conservative and may exclude low expressing trogocytic tumor cells, implying that the true percentage of tumor cells that are trogocytic may be much higher than this analysis estimates. This could account for the difference between the IF quantification and flow cytometry results.

## Cell culture

Human primary T cells were obtained from the Elutriation Core Facility at the University of Nebraska Medical Center. T cell populations were isolated using the Miltenyi Biotec Pan T Cell Isolation Kit (130-096-535). T cells were cultured in X-VIVO™ 15 Serum-Free Hematopoietic Cell Medium (Lonza, 02-053Q) supplemented with 20ng/mL recombinant human IL-2 (R&D, BT-002–010) once a week and 7ul/mL ImmunoCult™ Human CD3/CD28/CD2 T Cell Activator (Stem Cell Technologies, 10970) every other week. RCC cell lines (Caki-1, ACHN, A498, 786O) were obtained from Dr. George Sutphin's laboratory at the University of Arizona. RCC cell lines were cultured based on the recommendations made by ATCC.

## Lentiviral transduction of primary T cells

GFP-H2B lentiviral vector was obtained from Dr. Ghassan Mouneimne at the University of Arizona. Lentivirus was generated in HEK293T cells, and the viral supernatant was harvested for primary T cell infection. T cells were spun for 30 minutes at 3000 rpm with the virus. Lentivirus was removed the following day after incubation and GFP+ T cells were isolated using FACS selection 2 weeks after transduction.

## Coculture assays

RCC lines (ACHN, A498, Caki-1, 786-O) were plated a minimum of 4 hours prior to coculture. T cells were added to the culture at a 1:1 ratio and incubated overnight. Cocultures were harvested for experimental purposes 16 hours post addition of the T cells.

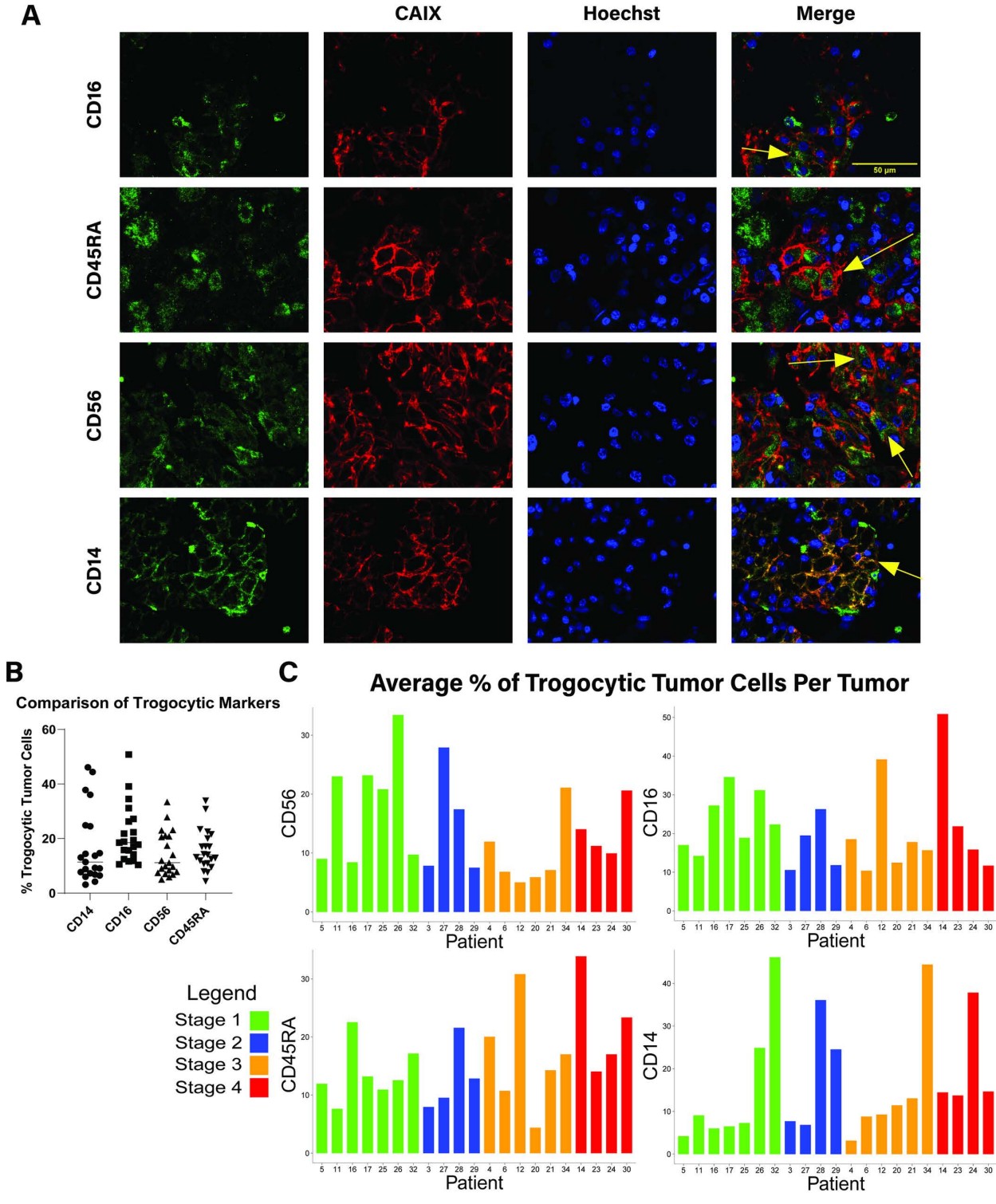

**Fig 1. Trogocytic RCC tumor cells are detected and quantified using immunofluorescent staining.** (A) Immunofluorescent microscopy of human RCC tumors. Trogocytic markers (CD14, CD56, CD45RA, CD16), CAIX, and Hoechst are displayed as green, red, and blue respectively. Arrows indicate trogocytic tumor cells. (B) Quantification of average percentage of trogocytic tumor cells for 21 human RCC tumors. (C) Quantification of average percentage of trogocytic tumor cells in 21 human RCC tumors organized by tumor stage. Methodology of quantification described in Methods and Materials.

### Confocal microscopy of EdU transfer

Figs 4 and 5 images were taken on a Zeiss 710 Laser Scanning Confocal microscope at 40X (NA = 1.30) or 63X (NA 1.40). Fig 4 images shown are max intensity projections of 60 Z-stacked images. FIJI smoothing algorithm was applied to phalloidin imaging for the purpose of minimizing the quality loss of actin labeling as a result of the EdU staining process.

### Ethics statement

These studies were determined to be exempt from IRB approval due to the use of deidentified human samples. Written consent for exemption was obtained from the University of Arizona eIRB.

## Results

### Identification and quantification of tumor cells presenting both tumor and lymphocyte markers in human RCC samples

Macrophages have been recently implicated in abating the effects of immune checkpoint inhibitors (ICI) in treating RCC through the process of trogocytosis with cancer cells [10]. Given that RCC is an immunogenic cancer with high levels of immune infiltration, we hypothesized that other types of immune cells may play an important role in shaping the tumor micro-environment through trogocytosis. We obtained paraffin embedded slides from 21 human RCC tumors of varying stages to determine whether markers of trogocytosis could be detected. Positive staining for carbonic anhydrase 9 (CAIX) was used to define tumor cells, as it is a well-characterized marker of hypoxia and is commonly upregulated in RCC cells [19].

To identify trogocytic tumor cells, slides were stained with CAIX and various immune markers of interest. This method of identification was chosen because there is currently a lack of predetermined markers that can be used to identify trogo-cytosis in cancer cells [16]. Fluorescent imaging revealed significant populations of cells displaying both CAIX and CD14, CD16, CD56, or CD45RA (Fig 1A). Interestingly, the localization of these immune markers within the tumor cells varied with some samples showing surface membrane localization and others showing cytosolic immune cell protein expression. While trogocytosis can explain the surface membrane localization of these proteins, other forms of contact mediated cargo exchange between cells, such as cell fusion, may be responsible for the internalization of immune cell proteins by cancer cells. Imaging of normal kidney tissue revealed no expression of these markers by healthy kidney epithelial cells (S1A Fig). These results revealed that macrophages, monocytes, NK cells, and CD45RA+ T cells are likely the most active participants of trogocytosis with RCC cells. The RA isoform of CD45 is typically expressed on naïve CD4+ and CD8+ T cells, until it is converted into the RO form upon the formation of memory cells [20]. This implies that naïve T cells could be more trogocytic than mature memory T cells in RCC.

To compare the percentage of tumor cells displaying signs of trogocytosis, we developed an image quantification program that records the percentage of individual cells' surface area occupied by a trogocytic marker based on IF images (S1B Fig). Cutoffs were established to define a tumor cell as trogo+ or trogo- based on what estimated percentage of their surface area was positive for a given immune marker. Through this analysis, we found that of the 21 patient tumor images, all displayed significant signs of trogocytosis between immune cells and tumor cells (Fig 1B, C). The average percentage of trogo+ RCC cells per tumor was between 15–20%, with some patient tumors as high as 50% trogo+ for certain markers.

### Characterization of trogocytic RCC tumors by flow cytometry

To demonstrate the transfer of CD45 transfer from a lymphocyte to a tumor cell *in vitro* we performed several coculture assays between RCC cell lines and Jurkat T cells or primary CD3+ T cells isolated from healthy donor PBMC. Cell lines were cocultured for 16 hours and then CD45 transfer was assessed via flow cytometry. Our results reveal that there was a significant increase in CD45 labeling on RCC cells that had been cocultured with primary T cells compared to RCC cells in mono-culture (Fig 2A, B). We also observed a significant increase in CD45 expression by RCC cell in cocultures done with Jurkat

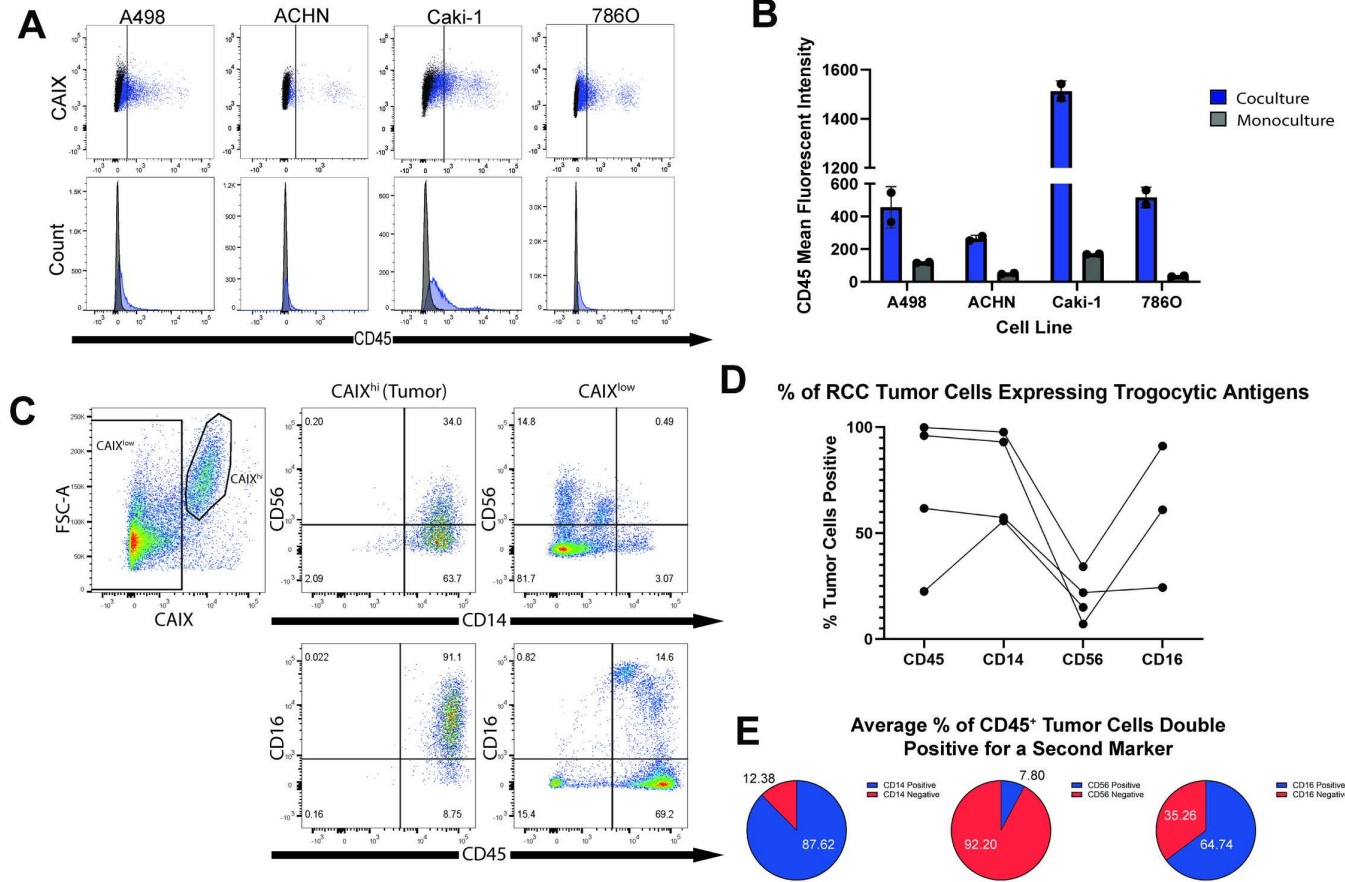

**Fig 2. RCC cells acquire expression of immune surface proteins through contact with lymphocytes.** (A) Flow cytometry analysis of CD45 expression by RCC cell lines post coculture with primary human T cells. RCC cells were cocultured with T cells for 16 hours prior to analysis. Data shows an increase in CD45 expression relative to RCC cells cultured in isolation (monoculture). (B) Mean fluorescent intensity of CD45 by cocultured RCC cells relative to monoculture. (C) Representative flow cytometry analysis of fresh human ccRCC tumors. Tumor cells were identified as CAIX^hi and lymphocytes and other cells present in the tumor microenvironment were identified as CAIX^low. Data shows expression of four immune cell surface proteins (CD14, CD16, CD56, CD45) by ccRCC and non-ccRCC cells isolated from this tumor. (D) Percentage of tumor cells from 4 human ccRCC tumors that expressed CD14, CD16 (n = 3 due to high autofluorescence in one sample), CD56, and CD45 based on flow cytometry analysis. Tumor cells were identified via CAIX^hi labeling as represented in Fig 2C. (E) Average % of CD45^+ tumor cells identified as double positive for a second trogocytosis marker. CD45^+ tumor cells were analyzed for expression of CD14 (n = 4, S.D. 5.66), CD56 (n = 4, S.D. 3.44), and CD16 (n = 3, S.D. 21.12).

T cells (S2A–C Fig). This transfer was inhibited when a transwell barrier was placed between the two cell types in coculture, indicating that physical contact is necessary (S2D–E Fig). We utilized size discrimination as well as CAIX expression to ensure no T cells were included when determining whether a CD45^+ cell was truly a trogocytic cancer cell (S2F–G Fig).

Next, we further characterized fresh human RCC tumors for markers of trogocytosis via flow cytometry. The IF analysis previously described has certain limitations that make distinguishing trogocytic staining from background signal difficult. Flow cytometry was chosen to provide a more accurate depiction of the trogocytic properties of RCC tumor cells. Fresh tumor samples were acquired from the University of Arizona biobanking facility and immediately dissociated and stained for markers of interest, namely CAIX, CD45, CD56, CD16 and CD14. Samples were evaluated within 12 hours of receiving them to ensure maximum viability of the cells.

Tumor cells were discriminated from infiltrating lymphocytes and non-tumor tissue by gating on CAIX^hi cells. Size discrimination was also used to determine that the CAIX^hi population was independent from lymphocyte populations present

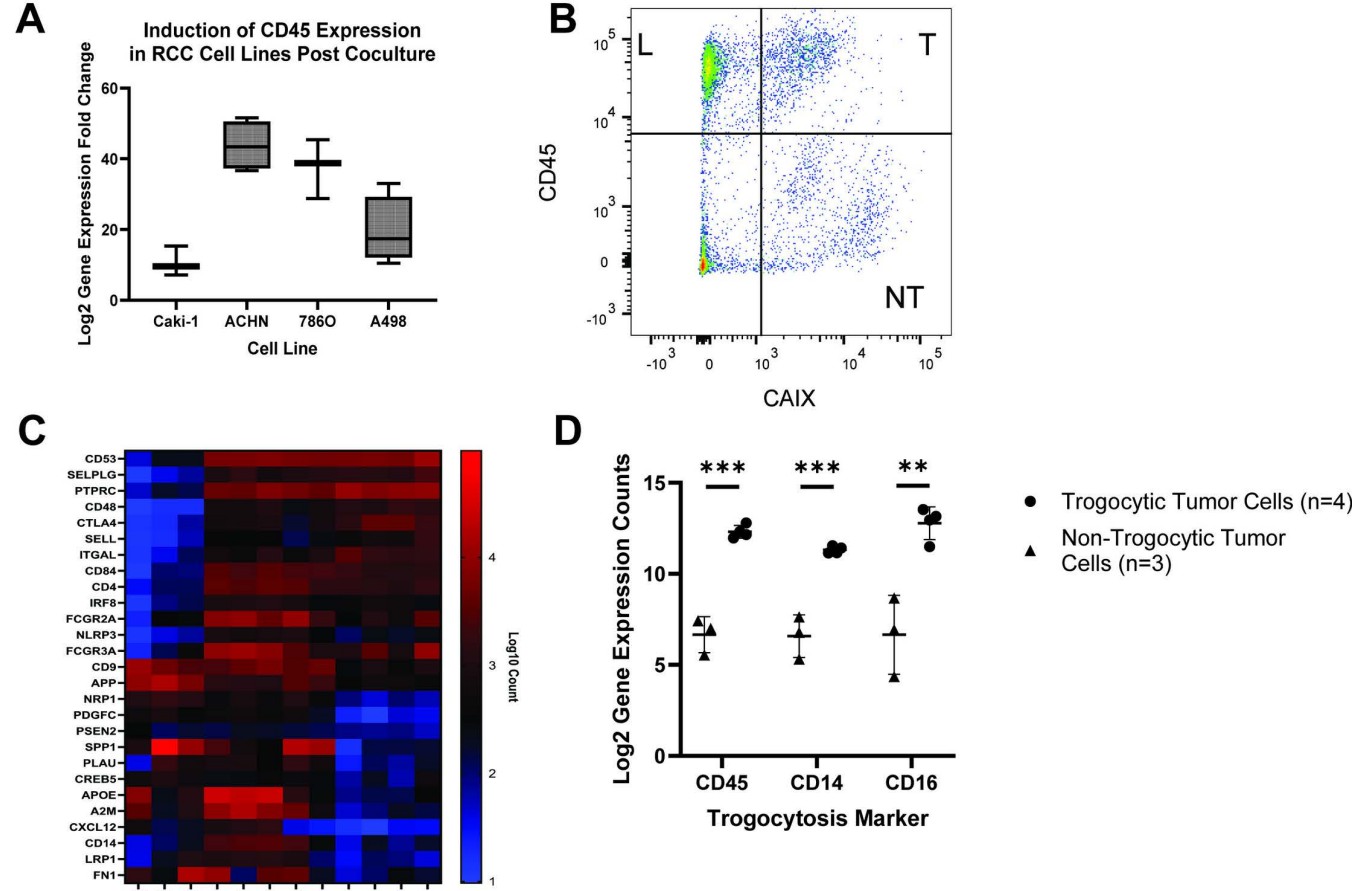

**Fig 3. RNA expression analysis of immune cell-specific genes by trogocytic RCC cells in human ccRCC tumors.** (A) qRT-PCR analysis of the change in CD45 expression in RCC cell lines post coculture with Jurkat T cells. Cells were sorted via FACS prior to analysis to isolate RCC cells. Graph depicts Log2 CD45 gene expression fold change in cocultured cells relative to monoculture. (Mean Ct values (n = 4): Caki-1 coculture 28.297, monoculture 31.174, ACHN coculture 29.131, monoculture 33.575, 786O coculture 28.501, monoculture 36.759, A498 coculture 29.071, monoculture 33.385). Respective p-values comparing fold change increase in CD45 expression after coculture as calculated by T-test: Caki-1 0.0050, ACHN 0.000019, 786O 0.00027, A498 0.0081. (B) Representative gating strategy used to identify lymphocyte (L), trogocytic tumor (T), and non-trogocytic tumor (NT) cell populations in human ccRCC tumors when sorting cells via FACS. L cells were identified as CD45high CAIX$^{low}$, trogocytic tumor cells were identified as CD45$^{high}$ CAIX$^{high}$, and non-trogocytic tumor cells were identified as CD45$^{low}$ CAIX$^{high}$. (C) NanoString® gene expression analysis for human ccRCC tumor. Populations were sorted from fresh human ccRCC tumors via FACS according to the gating strategy described in Fig 3A. Data shows Log10 RNA expression counts for select immune and cancer specific genes including CD45 (PTPRC) and CD16 (FCGR3A). (D) Comparison of Log2 gene expression counts of trogocytosis markers between trogocytic and non-trogocytic tumor cell populations isolated from human RCC tumors. Data was analyzed using multiple T tests (p value < 0.01 = **, p value < 0.001 = ***).

in the tumor (Fig 2C). After establishing the tumor cells as distinct from all other cell types present in the tissue, we further characterized both the tumor and lymphocyte populations using a panel of trogocytic markers (CD14, CD16, CD56, and CD45). Analysis showed significant levels of expression of all four markers by fresh human ccRCC tumor cells (Fig 2C, D and S3A, S5 Figs).

CD45 is an established marker for all nucleated cells of hematopoietic lineage [21]. CD45 labeling revealed widespread expression on the surface of the CAIX$^{hi}$ (tumor) population (Fig 2C, D). Our data suggest that the majority of tumor cells in RCC tumors have expression of one or more immune markers, and that this property is detectable by flow cytometry.

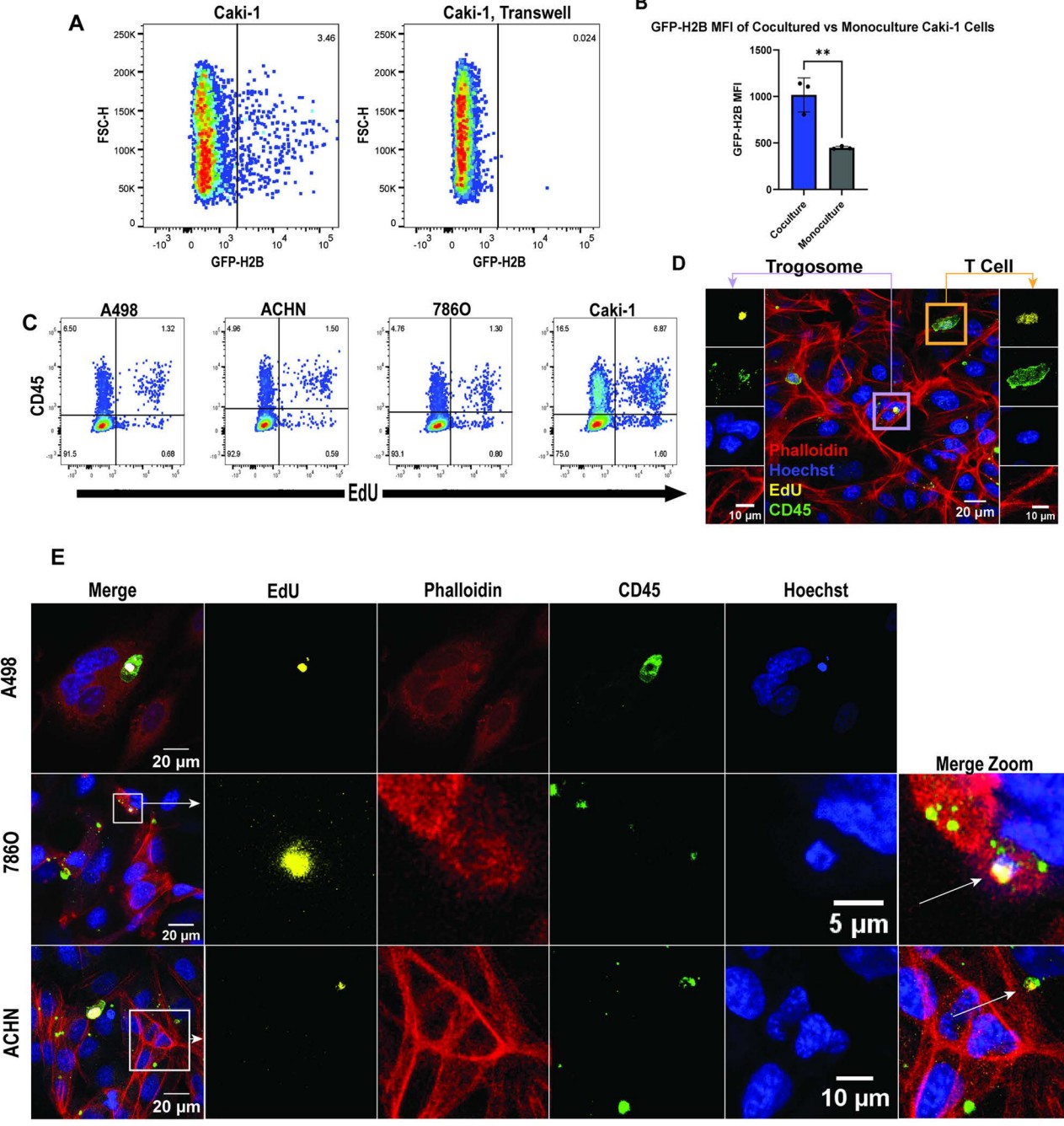

**Fig 4. DNA is transferred from primary T cells to RCC cell lines.** (A) Flow cytometry results of Caki-1 cells that were cocultured with GFP-histone H2B primary T cells. Data shows the percentage of GFP-H2B$^+$ Caki-1 cells in coculture relative to monocultured Caki-1 cells. (B) Mean fluorescent intensity of GFP-H2B expression of cocultured Caki-1 cells compared to monoculture (p value = 0.0058, unpaired T test was used to analyze the data). (C) Flow cytometry analysis of RCC cell lines cocultured with EdU-labeled primary human T cells. Plots show the percentage of RCC cells that are EdU$^+$ and CD45$^+$ post 16-hour coculture. (D) Max intensity projection of Z-stacked images taken from a 10:1 coculture of EdU-labeled primary human T cells with ACHN cells. Left panels depict an enhanced view of an EdU$^+$ CD45$^+$ trogosome within the intracellular space of an ACHN cell. Right panels depict an enhanced view of an EdU$^+$ CD45$^+$ T cell for comparison purposes. (E) Representative max intensity projection images of trogosomes in three RCC cell lines generated from Z-stacks (top to bottom, A498, 786O, ACHN). Boxes in the merged images of middle and bottom panels indicate the region being enhanced in the split color panels to the right of the merged image. Arrows in the merged zoom panels indicate the location of a trogosome. All images taken on a Zeiss 710 laser scanning confocal, 40X (NA 1.30).

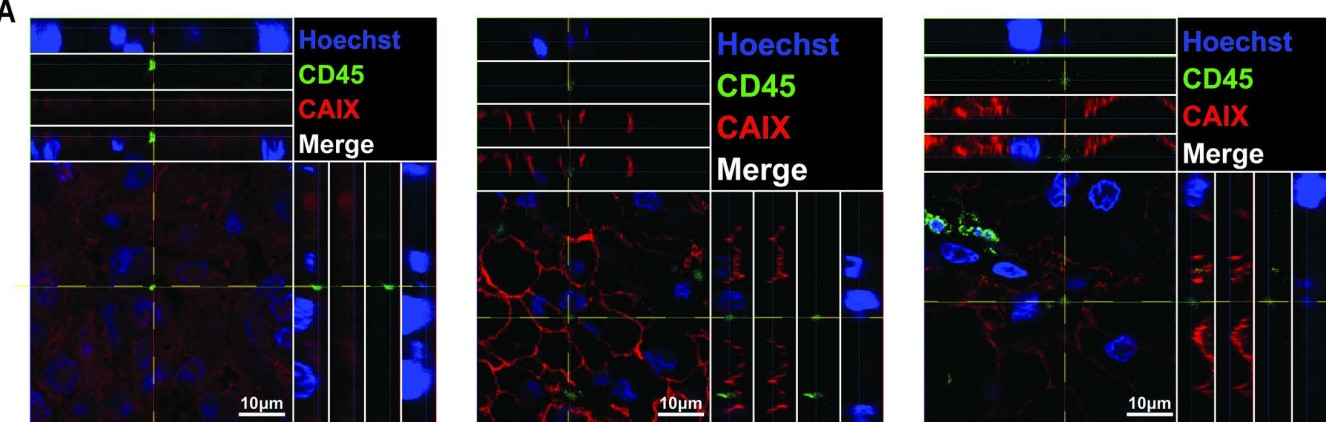

**Fig 5. Trogosomes identified in human RCC tumor samples using confocal microscopy.** (A) Trogosomes were identified in three representative human RCC images as CD45⁺ Hoechst⁺ subcellular bodies. Z-stacks were taken at each site and represented through max intensity projections. Top and side panels depict the merged and individual channels as orthogonal views through the planes indicated. Images taken at 63X on a Zeiss 710 laser scanning confocal (NA 1.40).

Given the high percentage of tumor cells exhibiting trogocytic markers, we then decided to focus on characterizing cells that express multiple immune proteins. Our data revealed that the vast majority of CAIX^hi cells were positive for multiple immune cell markers (Fig 2D). Therefore, it is possible that RCC tumor cells undergo multiple types of contacted mediated transfer with lymphocytes such as trogocytosis and cell fusion that allow them to "steal" the ability to express these markers from multiple types of immune cells. This is illustrated by analysis of double positive subpopulations of CD45⁺ tumor cells. On average, the majority of CD45⁺ cancer cells were also CD14⁺ (87.62%) and CD16⁺ (64.74%), with a minor fraction also expressing CD56 (7.80%) (Fig 2E). The exact identity of the immune cells that undergo trogocytosis with RCC cells is not yet fully elucidated and is an important field of study for future investigations.

## RCC tumor cells express RNA for immune surface proteins

Given the abundance of immune antigens being expressed by RCC tumor cells, we decided to identify the source of this protein expression. Upwards of 98% of tumor cells in human RCC tumors were positive for one or more trogocytic protein when characterized by flow cytometry (Fig 2C). The current definition of trogocytosis describes the transfer of membrane fragments from one cell to another, however it seemed unlikely that simple membrane transfer from lymphocytes to tumor cells could account for the relatively high quantities of trogocytic protein present on the surface of RCC cells. Therefore, we sought to identify other possible sources of this protein expression by tumor cells. More specifically, we attempted to determine whether tumor cells were capable of self-sufficient transcription of immune cell-specific proteins after contact with immune cells occurred. To investigate this, we cocultured Jurkat T cells with four kidney carcinoma cell lines. We performed fluorescence activated cell sorting (FACS) to isolate RCC cells from the coculture prior to harvesting RNA to ensure there was no contamination of T cells in the cells being analyzed (S4A Fig). Doublets were also discriminated against during cell sorting to ensure that immune cells engulfed by RCC cells were not included in the sorted population (S4B Fig). Using qRT-PCR analysis, we identified a significant induction of CD45 gene expression in RCC cells post coculture (Fig 3A).

Next, we isolated trogocytic (T, CAIX^hiCD45^hi), non-trogocytic (NT, CAIX^hiCD45^low), and lymphocyte (L, CAIX^lowCD45^hi) populations from human RCC tumor samples via FACS sterile sorting (Fig 3B). We then ran an nCounter® PanCancer Immune Profiling analysis on the sorted cells using Nanostring technology to identify key transcriptional differences between

T and NT cells. Our analysis showed that trogocytic tumor cells exhibit a "fusion" phenotype, defined as expressing both tumor and immune-specific genes (Fig 3C). There was significantly higher expression of the trogocytic markers CD45, CD14, and CD16 in trogocytic tumor cell populations compared to non-trogocytic tumor cell populations (Fig 3D). We also identified new potential trogocytic markers such as CD4 and CD53 on the trogocytic tumor samples (Fig 3B, S1 Table).

**RCC cell lines acquire genomic DNA from primary T cells**

In order to determine the mechanism by which RCC tumor cells gained immune protein expression, we investigated the possibility of horizontal gene transfer (HGT) occurring during cell-cell contact. To do this, we generated a primary T cell line expressing GFP-tagged histone H2B as a method of tracking DNA transfer. GFP-H2B T cells were cocultured overnight with Caki-1 cells and then analyzed via flow cytometry. Analysis showed that there was a small population of GFP-positive Caki-1 cells that was not present in monocultured Caki-1 cells (Fig 4A). We observed a significant increase in the mean fluorescence intensity of GFP in cocultured Caki-1 cells compared to monocultures of Caki-1 cells alone (Fig 4B).

Additionally, we isolated primary human T cells from a healthy donor and incubated them with EdU, a thymidine analog that is readily incorporated into the DNA of dividing cells [22]. EdU-labeled T cells were then cocultured overnight with RCC cell lines (ACHN, A498, Caki-1, and 786O cells). Following coculture, the cells were then analyzed via flow cytometry to determine whether we could detect a transfer of EdU-labeled DNA from the T cells to the RCC cell lines. Results showed that between 1–8% of cocultured RCC cells contained EdU labeled DNA (Fig 4C). As contamination from EdU labeled T cells could result in false positive results, we used size discrimination in the flow cytometry analyses to distinguish cancer cells from T cells (S6A,B Fig). In contrast, monocultured RCC cell lines and cocultures done with a transwell control separating the T cells from the RCC cells did not show any evidence of DNA transfer (S6C Fig).

Finally, we performed super-resolution structured illumination microscopy to gain insight into the physical properties of the transferred material. We discovered CD45+ intracellular bodies present in RCC cells post coculture with primary human T cells (Fig 4D, E). The appearance of these structures ranged from whole cell engulfment (Fig 4E, A498) to small fragment uptake (Fig 4D, E, 786O, ACHN). Processes such as emperipolesis, which is the engulfment of hematopoietic derived cells, may explain some of these observations [23]. Additionally, some of these "trogosomes" contained EdU labeled DNA (Fig 4D, E). Interestingly, we observed instances in which there was a lack of colocalization between EdU and Hoechst DNA labeling (Fig 4E, ACHN). This is likely due to the covalent attachment of bulky chemical groups to DNA as a result of EdU labeling that can block Hoechst from binding, particularly in A-T rich regions of DNA [22]. The presence of these trogosomes could be indicative of a mechanism by which lymphocyte DNA is transferred through CD45+ lymphocyte membrane bubbles into the intracellular space of RCC cells.

**CD45 positive trogosomes are observed in the intracellular space of ccRCC cancer cells in human tumors**

To determine the clinical relevance of lymphocyte DNA transfer to renal cancer cells through CD45+ trogosomes, we performed immunofluorescent imaging and analysis of human ccRCC tumors. Using confocal microscopy, we identified CAIX+CD45+ cells within these tumors (Fig 5A). Within the intracellular space of these tumor cells, we observed structures similar to the trogosomes that were identified in our *in vitro* coculture experiments. To assess the DNA content of these structures and confirm their presence within the cytosol of RCC cells, we generated orthogonal views along the X and Y planes containing the trogosome. This revealed that these CD45+ membrane bubbles contained extranuclear DNA within the cytoplasm of renal carcinoma cells (Fig 5A). Taken together, this data demonstrates that DNA transfer from immune cells to cancer cells occurs in human renal carcinoma tumors.

## Discussion

In this study, we showed that RCC tumor cells express significant levels of lymphoid proteins, such as CD45RA, CD14, CD16, and CD56. We have demonstrated that this expression is obtained through trogocytosis and subsequent horizontal

gene transfer occurring between a tumor cell and lymphocyte. Importantly, this phenomenon is widespread, with over 90% of fresh RCC tumor cells showing high levels of trogocytic protein expression when analyzed via flow cytometry. Additionally, RNA extraction and analysis showed that tumor cells exhibit an altered transcriptome after trogocytosis occurs. Namely, we observed an increase in RNA expression for genes that are typically only expressed by cells of hematopoietic lineage. Finally, we demonstrated that genomic DNA can be transferred during trogocytosis from a lymphocyte to a tumor cell resulting in an altered tumor cell phenotype. Taken together, this data demonstrates that RCC tumor cells acquire the ability to express large quantities of hematopoietic surface antigens following a trogocytic event with infiltrating lymphocytes.

Using immunofluorescent analysis of RCC slides, we developed a computer algorithm to quantify the percentage of tumor cells that express proteins found on lymphocytes. This study revealed consistent expression of the markers CD14, CD16, CD56, and CD45RA in a cohort of 21 human patient tumors when analyzed via immunofluorescence microscopy (IF). Despite limitations in the sensitivity of this method, analyzing 21 tumors originating from different stages of disease revealed no significant difference in trogocytic antigen expression at any stage of tumor development. Additionally, of the four trogocytic antigens we analyzed, there were no significant changes in the specific type of antigen being expressed at any stage. These data imply that trogocytosis occurs at an early stage during tumor development, and that the trogocytic phenotype of RCC tumor cells are maintained as disease progresses. Currently, it is unknown whether this is due to repeated trogocytic interactions with infiltrating lymphocytes, acquisition of lymphocytic DNA that promotes expression of these antigens through cell fusion, or some combination of the two.

Analysis of fresh tumors by flow cytometry corroborated the results we obtained via IF. Flow cytometry revealed much higher levels of trogocytic marker expression than previously anticipated, with over 90% of tumor cells in all RCC tumors tested being positive for at least one lymphocytic marker. To our knowledge, this is the first time these lymphocytic proteins have been identified on any solid tumor and could shift the paradigm of how we understand a tumor's interactions with its microenvironment.

Our investigation into horizontal gene transfer between RCC cells and T cells has shown that it is possible for tumor cells to acquire DNA from lymphocytes that they encounter. We have shown that the transfer of both EdU-labeled DNA and GFP-tagged histones is observed after coculturing tumor cells and immune cells *in vitro*. Currently, it is unknown whether this transfer confers greater survival capabilities or a more aggressive phenotype to tumor cells that have undergone horizontal gene transfer. Several groups have shown that cancer cells that undergo cell fusion can display cancer stem-cell-like qualities, meaning this process may act as an avenue to more severe disease [24,25]. Additional studies are needed to fully characterize this process and its role in disease progression. Our data also suggests that membrane transfer can occur independently of horizontal gene transfer, implying that there are multiple mechanisms that regulate these processes.

Trogocytosis has been shown to promote the acquisition of the immune check point receptor PD-1 by natural killer cells in mouse models of leukemia, resulting in suppressed NK cells antitumor immunity [26]. Additionally, colorectal cancer cells have been shown to obtain other immune regulatory molecules such as CTLA4 through trogocytosis with infiltrating lymphocytes [8]. These results suggest that trogocytosis is a mechanism by which tumors evade elimination by the immune system, either by acquiring regulatory molecules themselves, or by indirectly benefiting from certain immune cells negatively regulating themselves. In the context of RCC, it is unknown how tumor cells use the acquisition of immune cell surface protein and DNA to promote their survival. However, given the frequent expression of these markers on RCC tumors from all stages of disease, we believe trogocytosis and other contact dependent cell cargo transfer mechanisms like cell fusion confer a survival advantage to tumor cells. More research is needed to determine whether this survival advantage is selected for by the tumor to evade the immune system, or some other unknown mechanism.

It is possible that trogocytosis and the effect it has on the phenotype of tumor cells may be exploited to improve cancer therapies. The expression of lymphoid antigens on cancer cells may constitute effective targets of antibody drug

conjugates (ADCs). Further profiling of the markers that are transferred during this process may reveal unique protein expression signatures by trogocytic tumors that are not otherwise found in healthy cells. In addition to their use as new drug targets, these markers can also potentially be used as novel biomarkers to help better inform treatment strategies for clear cell renal carcinoma patients and possibly other types of solid tumors. Further research in the field of trogocytosis is critical for developing the next generation of immunotherapies.

## Supporting information

**S1 Fig. (A) Immunofluorescent staining of normal kidney tissue.** Trogocytic markers shown in the left panel, ZO-1 staining highlights normal kidney epithelial cells. (B) Representative illustration of the algorithm used to quantify immunofluorescent kidney slides. (B,I) Cancer cells represented in red (CAIX), lymphocytes represented in green (CD45). Trogocytic tumor cells represented by expression of both CAIX and CD45. (B,II) Detection of nuclei through hoechst staining, Voronoi diagram is applied based on the location of nuclei and segments cells. (B,III and IV) CAIX staining is applied to the previous voronoi diagram and used to exclude non-CAIX$^+$ cells such as lymphocytes and non-tumor tissue. (B,V) Final voronoi diagram is applied to CD45 labeling to approximate the boundaries of cells and determine the expression levels of CD45 on tumor cells only.
(TIF)

**S1 Table. Top 20 differentially expressed genes detected in the Nanostring analysis of fresh ccRCC tumors.** (Left) Results show a comparison of the most differentially upregulated genes in trogocytic tumor cells relative to non-trogocytic tumor cells based on the PanCancer Immune Profiling Nanostring® panel. (Right) Top 20 differentially upregulated genes in trogocytic tumor cells relative to tumor infiltrating lymphocytes. Results are based on mean gene expression counts of 3 non-trogocytic tumor cell populations, 4 trogocytic tumor cell populations, and 5 tumor infiltrating lymphocyte populations isolated from human ccRCC tumors.
(TIF)

**S2 Fig. (A-C) Flow cytometry analysis of RCC cell lines that were cocultured with Jurkat T cells to asses the transfer of CD45.** (Top) Gating indicates the percentage of RCC cells that are CD45$^+$ relative to monoculture controls. (D,E) Flow cytometry analysis of A498 and 786O cocultures with primary human T cells that were separated by a transwell barrier. Gating indicated the percentage of RCC cells that were positive for CD45 post-over night coculture. (F-H) Flow cytometry gating strategy used to remove T cell for analysis of CD45 positive cancer cells. Example depicts A498 cells cocultured with primary T cells (F), A498 cells only (G), and A498 cells separated from primary T cells using a transwell barrier (H). (I) Representative plot of CAIX transfer from tumor cells to T cells post coculture with Caki-1 cells. (J) Quantitative comparison of %CAIX$^+$ T cells post coculture with Caki-1 cells showing no significant transfer of CAIX (T-test, p value = 0.135).
(TIF)

**S3 Fig. Flow cytometry results of three additional fresh human RCC tumors.** Left plots indicate the gating strategy used to isolate CAIX$^+$ tumor cells and CAIX$^{low}$ lymphocytes. The right plots for each tumor indicate the percentage of cells in these two populations that are positive for the indicated immune cell proteins.
(TIF)

**S4 Fig. FACS sorting strategy for qPCR analysis of cocultures.** (A) Boxes indicate cells that were sorted for RNA purification. (B) Doublet discrimination of cells prior to sorting gate. Boxes indicate the location of singlets, red arrows indicate the location of doublets that were not included in the sorts.
(TIF)

**S5 Fig.  Multiparametric representation of an additional fresh kidney tumor flow cytometry data based on expression of the trogocytosis markers CD45, CD14, CD16, CD56, and CD68.** Prior to analysis, tumor cells were isolated based on CAIX$^+$ expression, and size discrimination.
(TIF)

**S6 Fig.  Comprehensive gating strategy used to identify EdU positive RCC cells (A498 depicted).** (A) Single cells in coculture suspension were identified using FSC-H FSC-A discrimination. Live cells were determined using UV Ghost Dye 450 as a marker for viability. SSC-A and FSC-A were used for identifying T cells and RCC cells, subsequently isolating RCC cells only for analysis. Red arrow indicates the position of T cells. FSC-A and EdU plot depicts the gate used to identify EdU positive cancer cells. (B) Gating strategy applied to a monoculture of only A498 cells. (C) Gating strategy applied to an A498-T cell coculture that was separated by a transwell barrier. T cells were removed with the barrier prior to flow cytometry analysis.
(TIF)

## Acknowledgments

Major instrumentation has been provided by the Office of the Vice Chancellor for Research, The University of Nebraska Foundation, the Nebraska Banker's Fund, and by the NIH-NCRR Shared Instrument Program. Special thanks to J.C. Fitch for assistance with flow cytometry analysis.

## Author contributions

**Conceptualization:** Haley Q. Marcarian, Alfred Lester Meador Bothwell.

**Data curation:** Haley Q. Marcarian, Anika M. Arias, Olivia C. Ihedioha.

**Formal analysis:** Haley Q. Marcarian.

**Funding acquisition:** Alfred Lester Meador Bothwell.

**Investigation:** Haley Q. Marcarian, Anutr Sivakoses, Anika M. Arias, Olivia C. Ihedioha, Alfred Lester Meador Bothwell.

**Methodology:** Haley Q. Marcarian, Anutr Sivakoses, Anika M. Arias.

**Project administration:** Alfred Lester Meador Bothwell.

**Resources:** Benjamin R. Lee, Alfred Lester Meador Bothwell.

**Software:** Haley Q. Marcarian.

**Supervision:** Benjamin R. Lee, Maria C. Bishop, Alfred Lester Meador Bothwell.

**Validation:** Haley Q. Marcarian.

**Visualization:** Haley Q. Marcarian.

**Writing – original draft:** Haley Q. Marcarian.

**Writing – review & editing:** Anutr Sivakoses, Olivia C. Ihedioha, Alfred Lester Meador Bothwell.

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
