## [Decision Letter · Decision Letter 0]

26 Mar 2025

PONE-D-24-53647Renal cancer cells acquire immune surface protein through trogocytosis and horizontal gene transferPLOS ONE

Dear Dr. Bothwell,

Thank you for submitting your manuscript to PLOS ONE. After careful consideration, we feel that it has merit but does not fully meet PLOS ONE’s publication criteria as it currently stands. Therefore, we invite you to submit a revised version of the manuscript that addresses the points raised during the review process.

We look forward to receiving your revised manuscript.

Kind regards,

Kenji Fujiwara, MD, PhD, FACS

Academic Editor

PLOS ONE

 [Studies were supported by funding from the National Cancer Institute (T32CA009213-44) to A. Sivakoses and 7R01AI137060-06  awarded to A.L.M. Bothwell.

https://www.cancer.gov/]. 

5.  We are unable to open your Figures file [Fig1.eps, Fig2.eps, Fig3.eps, Fig4.eps and Fig5.eps]. Please kindly revise as necessary and re-upload.

Additional Editor Comments:

Dear Dr. Bothwell.

The article was reviewed by two reviewers and both recommended major revisions. Please check the comments carefully and send the revised one in time-dependent manner.

Best regards,

Kenji Fujiwara

Academic editor

Reviewers' comments:

Reviewer's Responses to Questions

**Comments to the Author**

1. Is the manuscript technically sound, and do the data support the conclusions?

Reviewer #1: Partly

Reviewer #2: Partly

2. Has the statistical analysis been performed appropriately and rigorously? 

Reviewer #1: Yes

Reviewer #2: Yes

3. Have the authors made all data underlying the findings in their manuscript fully available?

Reviewer #1: Yes

Reviewer #2: Yes

4. Is the manuscript presented in an intelligible fashion and written in standard English?

Reviewer #1: Yes

Reviewer #2: Yes

5. Review Comments to the Author

Reviewer #1: Marcarian et al. discovered that human clear cell renal carcinoma can gain immune cell markers through both trogocytosis and horizontal gene transfer. Although trogocytosis was reported in several previous research, most of them focus on antigen transfer from tumor cells to immune cells. This finding provides a novel perspective in the field of cancer immunology and tumor microenvironment. However, the molecular mechanism of trogocytosis and how tumor benefit from gaining immune cell markers need further studied. I believe after moderate revision experiments and polishing to the text and figures, this could be a good fit for PLOS ONE.

Major concerns:

1. Trogocytosis discovered by previous studies were mediated by some level of interaction of the surface molecules between the two cells (TCR to MHC-peptide complex; CAR-antigen; CTLA4-B7 complex). What interacting molecules are responsible for trogocytosis of immune cells markers to tumor cells? Based on the result, trogocytosis discovered by the authors are stochastic and random.

2. To what extend this type of trogocytosis rely on the nature/characteristic of immune cells? Would you still observe trogocytosis if you coculture two tumor lines, with one of them overexpress immune cell marker like CD45?

3. Previous studies have shown that trogocytosis happens in both directions. Is this type of trogocytosis discovered by the authors also reciprocal (meaning that tumor markers like CA9 also found transferred to immune cells?

Minor concerns:

1. Figures should be citated/callout sequentially. However, Figure 3C was called out before Figure 3A and 3B.

2. Figure 2D:

a. N = 4 in the legend but there are only three data points for CD16. Please explain where the missing data point goes.

b. This figure is not very good at demonstrating the claim made in the result section: “Our data revealed that the vast majority of CAIXhi cells were positive for multiple immune cell markers.” It’s better to use one or multiple bar plots or pie charts to show the percent of quadrupole, triple, or double positive cells for each sample. For example, 30% CD45 positive and 30% CD14 positive doesn’t necessary mean 30% of the cells are double positive for these two markers. Very likely these 30% were not the same group of cells.

c. It would be better if use different symbol for different samples (or connected same sample with line). With this additional information, it would be interesting to see whether some sample have higher trogocytosis for all markers consistently.

3. Figure 3A only shows gating strategy for Lymphocytes, Non_trogocytic tumor, and Trogocytic tumor. It can be moved to supplemental.

4. Figure 3B Nanostring data would be better to include more immune markers. For example, Figure 3C and figure 3D both show upregulation of makers like CD45, CD14, and CD16.

5. The authors should cite more current research articles in the introduction part. Recommended high impact paper to cite:

a. Huang, J. F. et al. TCR-Mediated internalization of peptide-MHC complexes acquired by T cells. Science 286, 952–954 (1999).

b. Hamieh, M. et al. CAR T cell trogocytosis and cooperative killing regulate tumour antigen escape. Nature 568, 112–116 (2019).

c. Zhou, X., Cao, H., Fang, SY. et al. CTLA-4 tail fusion enhances CAR-T antitumor immunity. Nat Immunol 24, 1499–1510 (2023).

d. Schriek, P. et al. Marginal zone B cells acquire dendritic cell functions by trogocytosis. Science 375, eabf7470 (2022).

Reviewer #2: Comments to manuscript PONE-D-24-53647

In their present work the authors showed that renal cancer cells were positive for lymphocyte/leukocyte markers, such as CD45, CD56, CD14 and CD16. The authors assumed that this finding was attributed to trogocytosis and horizontal gene transfer, which would be a novel mechanism by which tumor cell protein expression is altered through the acquisition of surface membrane fragments and genomic DNA from infiltrating lymphocytes. In brief, this is an interesting finding, but the available data provide only limited support for the assumption that the process why lymphocyte/leukocyte markers can be found in renal cancer cells could be trogocytosis. Other mechanisms such as cell cannibalism, entosis and cell fusion must also be considered.

1) In addition to trogocytosis, other mechanisms have been identified that would explain the presence of lymphocyte/leukocyte proteins, such as CD45, CD56, CD14 and CD16 and lymphocyte/leukocyte DNA in cancer cells, which are cell cannibalism, entosis and cell fusion (e.g., Fais et al., 2018 Cell Death & Disease 9: 95; Kapsetaki et al., 2024 Scientific Reports 14: 7535; Shultes et al., 2024 The International Journal of Biochemistry & Cell Biology 175: 106649; Sieler et al., 2024 Results Probl Cell Differ 71:433). Thus, I would very much appreciate if the authors can thoroughly discuss what speaks for trogocytosis, cell cannibalism, entosis or cell fusion and what not.

As the authors know, trogocytosis is a process in which one cell physically extracts and ingests “bites” of cellular material from another cell (Bettadapur et al., 2020 doi.org/10.1128/iai.00930-19), but to my knowledge this cellular material rather comprises of parts of the plasma membrane and a certain cytosolic fraction, but not DNA. Thus, it remains unclear why leukocyte/lymphocyte DNA was found in renal cancer cells.

Cell fusion might be an explanation for this finding. In this regard, former studies of Yilmaz et al. and Rachkowsky et al. demonstrated that donor DNA was found in renal carcinoma cells of female BMT recipients (Bone Marrow Transplant 2005 Vol. 35 Issue 10 Pages 1021-4; Bone Marrow Transplant 2004 Vol. 34 Issue 2 Pages 183-6).

However, it cannot be ruled out that cancer cells have phagocytosed immune cells or that immune cells were taken up by cancer cells by entosis. For instance, in Fig. 4E A498 it seems that the A498 cell harbors a rather intact T-cell, which might be related to either cell cannibalism or entosis. However, it cannot be ruled out that this T-cell is simply attached on top of the A498 cell and not inside. Thus, 3D images with Z-stacks would be helpful. This also applies to the ACHN data in Fig. 4E. Unfortunately, the resolution is too low to conclude that the marked T cell is really inside the cancer cell or not. Nonetheless, the irregular form of the nucleus and the scattered CD45 staining pattern might indicate that lysis of the engulfed T-cell has already started, which would rather speak for cell cannibalism than for entosis. This might also be true for 768O data. Here, CD45 staining was not colocalized with EdU staining.

The assumption that rather cell cannibalism, entosis or cell fusion might be responsible for the presence of T-cell DNA in RCC cells (and possibly not trogocytosis) is further supported by data presented in Fig. 4A and 4C. As mentioned above, trogocytosis is rather the ingestion of small cellular parts of a donor cell, which usually did not contain DNA. Data presented in Fig. 4A and 4C indicate a partially high GFP-H2B/EdU fluorescence in single Caki-1 cells, which cannot be sufficiently explained by the uptake of small amounts of T-cell DNA. In this regard, it is more likely that whole GFP-H2B/EdU expressing/labeled T-cells were engulfed/ phagocytosed by Caki-1 cells. Moreover, appropriate FACS data for wildtype Jurkat cells and GFP-H2B Jurkat cells and Edu-stained Jurkat cells are missing.

2) In Fig 1A it is unclear why only CD14 was found in the membrane of renal cancer cells, but not CD16, CD45RA and CD56. CD16, CD45RA and CD56 are all membrane proteins and if parts of the macrophage membrane were actively transferred to the cancer cell membrane by trogocytosis one would expect that macrophage derived membrane protein would be found in the plasma membrane of the cancer cells. However, it cannot be ruled out that cancer cells have “eaten” small cellular parts of the macrophages, which would explain the cytosolic localization of CD16, CD45RA and CD56. But then CD14 should also be found in the cytosol of the cancer cells, which obviously is not seen here. Please explain.

3) The sentence “Using qRT-PCR analysis, we identified a significant induction of CD45 gene expression in RCC cells post coculture (Fig. 3C)” (line 203-204) should be revised as it suggests that RCC cells actively up-regulated CD45 expression. Instead, CD45 expression was rather attributed to the engulfed immune cells. By doing so, the heading “ RCC tumor cells express RNA for immune surface proteins” (line 189) must also be revised since it also implies that, e.g., CD14, etc. is expressed in RCC cells.

Similarly, statistical significance was not calculated. Thus, if the authors conclude that the observed higher CD45 expression was significant, this should be validated by an appropriate statistical test (like in Fig. 3D).

Moreover, Fig. 3C (line 204) was mentioned in the text before the Figure legend of Fig 3 (line 206 – 223) and Fig. 3A (line 227) and 3B (line 231). Please rearrange the order.

4) The description of nanostring analysis is missing in the Materials and Methods section. Please revise.

5) In Figure 4B monocultured Caki-1 cells showed a basal GFP-H2B fluorescence. I guess that this is the autofluorescenc/ background fluorescence of the cells.

6) Data presented in Fig. 5 also rather support the assumption that whole CD45+ cells were engulfed/ phagocytosed by renal cancer cells and that the presence of CD45+ cells was not attributed to trogocytosis. For instance, results in the first row likely indicate that an intact CD45+ immune cells was freshly engulfed/ phagocytosed. However, 3D images with Z-stacks would be more helpful to clarify whether these CD45+ immune cells were really engulfed/ phagocytosed by renal cancer cells. Data presented in the second and fourth row are less convincing. In the second row only a very faint Hoechst signal can be seen. Similarly, no and even not a faint Hoechst signal of the prospective CD45+ immune cell can be seen in the fourth row. Thus, it is difficult for me to conclude whether these Hoechst signals were really attributed to immune cell DNA in renal cancer cells. However, it cannot be ruled out that the weak Hoechst signal was lost due to data compression. Data presented in the third row also support the assumption that an intact CD45+ immune cells that was freshly engulfed/ phagocytosed by a renal cancer cell. However, the shape of the Hoechst and CD45 staining in Fig. 5B does not really fit with the appropriate data in Fig. 5A. It looks as if the picture has been rotated 90° to the left. For instance, a faint Hoechst and CD45 stain can be seen in the upper right corner, which, however, cannot be seen in the appropriate image of Fig. 5A. This must be checked and corrected.

7) Transmission images should be provided for all immunofluorescence data.

6. PLOS authors have the option to publish the peer review history of their article (what does this mean? ). If published, this will include your full peer review and any attached files.

**Do you want your identity to be public for this peer review?** For information about this choice, including consent withdrawal, please see our Privacy Policy .

Reviewer #1: No

Reviewer #2: **Yes: ** Thomas Dittmar

---

## [Author Response · Author response to Decision Letter 1]

21 Apr 2025

Dear editor and reviewers,

We would like to gratefully acknowledge the effort and care put forth by our academic editor and reviewers for the manuscript “Renal cancer cells acquire immune surface protein through trogocytosis and horizontal gene transfer”, Marcarian et al. We found the feedback received to be constructive and quite helpful in our efforts to strengthen our manuscript so that it may be suitable for publication. The points brought up by reviewers were insightful and we are grateful for their expertise. Here, we would like to address each point of concern raised by reviewers:

“Trogocytosis discovered by previous studies were mediated by some level of interaction of the surface molecules between the two cells (TCR to MHC-peptide complex; CAR-antigen; CTLA4-B7 complex). What interacting molecules are responsible for trogocytosis of immune cells markers to tumor cells? Based on the result, trogocytosis discovered by the authors are stochastic and random.”

Thank you for your feedback here. This point certainly raises interesting questions regarding what signaling pathway drives trogocytosis in renal carcinoma, however we feel that this falls beyond the scope of this manuscript. Our goal with this manuscript is to demonstrate that transfer of cellular components occurs between immune cells and tumor cells in human RCC tumors and in vitro. To define the mechanism behind this transfer would require a significant investment of time, as the signaling pathway that drives trogocytosis is currently unknown.

“To what extend this type of trogocytosis rely on the nature/characteristic of immune cells? Would you still observe trogocytosis if you coculture two tumor lines, with one of them overexpress immune cell marker like CD45?”

While we believe the question of what role immune cells play in trogocytosis is highly important for the field, we feel it falls beyond the scope of this manuscript. Our overall goal with this publication is to demonstrate that cellular materials are transferred to RCC cells and that this transfer affects gene transcription. Additionally, such lines are unavailable in RCC cell lines. We thank you for your feedback, however we feel the recommended experiment is not needed to support our claims.

“Previous studies have shown that trogocytosis happens in both directions. Is this type of trogocytosis discovered by the authors also reciprocal (meaning that tumor markers like CA9 also found transferred to immune cells?”

Thank you, this is an excellent point. We have added two additional panels to Supplemental Figure 2 showing there is not a significant amount of transfer of CAIX to T cells. However, this does not rule out the possibility that other proteins may be transferred. To date, the proteins that can be transferred during trogocytosis have not been catalogued. Investigating trogocytosis in the opposite direction (i.e. tumor cells to immune cells) is an important area of study and likely has important implications for the field.

“Figures should be citated/callout sequentially. However, Figure 3C was called out before Figure 3A and 3B.”

Figure 3 has been rearranged to reflect the order in which subpanels are called in the text.

“Figure 2D:

a. N = 4 in the legend but there are only three data points for CD16. Please explain where the missing data point goes.”

An explanation for the missing data point has been added in the figure legend.

“b. This figure is not very good at demonstrating the claim made in the result section: “Our data revealed that the vast majority of CAIXhi cells were positive for multiple immune cell markers.” It’s better to use one or multiple bar plots or pie charts to show the percent of quadrupole, triple, or double positive cells for each sample. For example, 30% CD45 positive and 30% CD14 positive doesn’t necessary mean 30% of the cells are double positive for these two markers. Very likely these 30% were not the same group of cells.”

Thank you for this feedback. Figure 2E has been added to provide additional data regarding the frequency of double positive cancer cells.

“c. It would be better if use different symbol for different samples (or connected same sample with line). With this additional information, it would be interesting to see whether some sample have higher trogocytosis for all markers consistently.”

This is a great suggestion, thank you. The bar graph has been changed to a line graph to better describe each individual sample in terms of the % positive cancer cells.

“3. Figure 3A only shows gating strategy for Lymphocytes, Non_trogocytic tumor, and Trogocytic tumor. It can be moved to supplemental.”

It is our preference to keep it in the main figure but thank you for the suggestion.

“4. Figure 3B Nanostring data would be better to include more immune markers. For example, Figure 3C and figure 3D both show upregulation of makers like CD45, CD14, and CD16.”

Thank you, the NanoString data has been expanded to include more immune markers including CD45, CD14, and CD16. We have also clarified the gene names for CD16 and CD45 in the figure legend so that it will be more clear what genes are present in the panel.

“5. The authors should cite more current research articles in the introduction part. Recommended high impact paper to cite:

a. Huang, J. F. et al. TCR-Mediated internalization of peptide-MHC complexes acquired by T cells. Science 286, 952–954 (1999).

b. Hamieh, M. et al. CAR T cell trogocytosis and cooperative killing regulate tumour antigen escape. Nature 568, 112–116 (2019).

c. Zhou, X., Cao, H., Fang, SY. et al. CTLA-4 tail fusion enhances CAR-T antitumor immunity. Nat Immunol 24, 1499–1510 (2023).

d. Schriek, P. et al. Marginal zone B cells acquire dendritic cell functions by trogocytosis. Science 375, eabf7470 (2022).”

Thank you for the suggestion. Articles have been added to the references.

“In addition to trogocytosis, other mechanisms have been identified that would explain the presence of lymphocyte/leukocyte proteins, such as CD45, CD56, CD14 and CD16 and lymphocyte/leukocyte DNA in cancer cells, which are cell cannibalism, entosis and cell fusion Thus, I would very much appreciate if the authors can thoroughly discuss what speaks for trogocytosis, cell cannibalism, entosis or cell fusion and what not.”

Thank you for providing this perspective. We have included more discussion of cell fusion in our text, particularly regarding the transfer of DNA from one cell to another (line 79-85). We have made efforts to rewrite certain sections so that it is clear trogocytosis is a separate process from the transfer of genomic DNA and this transfer is likely due to a previously identified cell-cell interaction like cell fusion.

“However, it cannot be ruled out that cancer cells have phagocytosed immune cells or that immune cells were taken up by cancer cells by entosis. For instance, in Fig. 4E A498 it seems that the A498 cell harbors a rather intact T-cell, which might be related to either cell cannibalism or entosis. However, it cannot be ruled out that this T-cell is simply attached on top of the A498 cell and not inside. Thus, 3D images with Z-stacks would be helpful. This also applies to the ACHN data in Fig. 4E. Unfortunately, the resolution is too low to conclude that the marked T cell is really inside the cancer cell or not. Nonetheless, the irregular form of the nucleus and the scattered CD45 staining pattern might indicate that lysis of the engulfed T-cell has already started, which would rather speak for cell cannibalism than for entosis. This might also be true for 768O data. Here, CD45 staining was not colocalized with EdU staining.” The assumption that rather cell cannibalism, entosis or cell fusion might be responsible for the presence of T-cell DNA in RCC cells (and possibly not trogocytosis) is further supported by data presented in Fig. 4A and 4C. As mentioned above, trogocytosis is rather the ingestion of small cellular parts of a donor cell, which usually did not contain DNA. Data presented in Fig. 4A and 4C indicate a partially high GFP-H2B/EdU fluorescence in single Caki-1 cells, which cannot be sufficiently explained by the uptake of small amounts of T-cell DNA. In this regard, it is more likely that whole GFP-H2B/EdU expressing/labeled T-cells were engulfed/ phagocytosed by Caki-1 cells. Moreover, appropriate FACS data for wildtype Jurkat cells and GFP-H2B Jurkat cells and Edu-stained Jurkat cells are missing.

Thank you for this feedback. We have clarified that the images in Figure 4 were generated from Z-stacked images that were taken within the boundaries of the RCC cells. Confocal microscopy was used here to ensure that out of focus light (such as light coming from objects outside the RCC cell membrane) would not be imaged. Additionally, we have included a discussion about possible cell-cell interactions that may be responsible for this transfer (lines 410-413). In general, we have tried to take a more agnostic approach in terms of what exact mechanism is responsible for this transfer.

“2) In Fig 1A it is unclear why only CD14 was found in the membrane of renal cancer cells, but not CD16, CD45RA and CD56. CD16, CD45RA and CD56 are all membrane proteins and if parts of the macrophage membrane were actively transferred to the cancer cell membrane by trogocytosis one would expect that macrophage derived membrane protein would be found in the plasma membrane of the cancer cells. However, it cannot be ruled out that cancer cells have “eaten” small cellular parts of the macrophages, which would explain the cytosolic localization of CD16, CD45RA and CD56. But then CD14 should also be found in the cytosol of the cancer cells, which obviously is not seen here. Please explain.”

We agree with the possibility that the cytosolic localization of these proteins is due to a process that is not trogocytosis. We have included a brief discussion clarifying the definition of trogocytosis (surface membrane fragment transfer) and how that differs from other mechanisms of transfer between cells such as cell fusion. We believe our data supports the notion that the complexity of the tumor microenvironment leads to multiple mechanisms of cellular transfer occurring. This explains the differences in immune protein localization in the cancer cells when observed via immunofluorescence.

It is worth noting that all four markers can be found localized to the surface membrane of these tumor cells when analyzed by flow cytometry. As we note in line 191, there are inherent limitations to the sensitivity of IF, particularly in complex samples like tumor tissue. This can make it difficult to depict faint surface membrane staining on samples that have high amounts of a protein of interest in the cytosol (or elsewhere in the frame for that matter).

3) The sentence “Using qRT-PCR analysis, we identified a significant induction of CD45 gene expression in RCC cells post coculture (Fig. 3C)” (line 203-204) should be revised as it suggests that RCC cells actively up-regulated CD45 expression. Instead, CD45 expression was rather attributed to the engulfed immune cells. By doing so, the heading “ RCC tumor cells express RNA for immune surface proteins” (line 189) must also be revised since it also implies that, e.g., CD14, etc. is expressed in RCC cells.

To eliminate the possibility of a “cell-in-cell” body being included in the sorted populations that were analyzed in our qPCR experiments, FSC-A and FSC-H discrimination of doublet cells was done during cell sorting. We have added an additional supplemental figure showing this gating strategy for all four RCC lines that were used (S4 A,B). This data, when considered alongside the NanoString analysis of fresh tumors, demonstrates that transcription of the cancer cells is affected after contact with a lymphocyte. The in vitro coculture between the four RCC lines and Jurkat cells was done as a proof of concept to validate the idea that the “trogocytic” tumor cells analyzed with NanoString displayed such a phenotype as a result of their transcription being altered through this contact. We do not deny the notion of immune cell engulfment occurring in our coculture. Rather, we think doublet discrimination during cell sorting is sufficient to rule this out as a possibility in these experiments.

Similarly, statistical significance was not calculated. Thus, if the authors conclude that the observed higher CD45 expression was significant, this should be validated by an appropriate statistical test (like in Fig. 3D).

We have performed the relevant statistical calculations for this panel. Resulting p-values are reported in the figure legend.

“Fig. 3C (line 204) was mentioned in the text before the Figure legend of Fig 3 (line 206 – 223) and Fig. 3A (line 227) and 3B (line 231). Please rearrange the order.”

Figure 3 has been rearranged to fit the order in which subpanels are called in the text.

“4) The description of nanostring analysis is missing in the Materials and Methods section. Please revise.”

We have added an additional section describing the NanoString analysis and procedure to the Methods and Materials section.

“5) In Figure 4B monocultured Caki-1 cells showed a basal GFP-H2B fluorescence. I guess that this is the autofluorescenc/ background fluorescence of the cells.”

This is correct, Caki-1 cells exhibit a certain level of autofluorescence which is reflected in both Figure 4A and B.

“6) Data presented in Fig. 5 also rather support the assumption that whole CD45+ cells were engulfed/ phagocytosed by renal cancer cells and that the presence of CD45+ cells was not attributed to trogocytosis. For instance, results in the first row likely indicate that an intact CD45+ immune cells was freshly engulfed/ phagocytosed. However, 3D images with Z-stacks would be more helpful to clarify whether these CD45+ immune cells were really engulfed/ phagocytosed by renal cancer cells. Data presented in the second and fourth row are less convincing. In the second row only a very faint Hoechst signal can be seen. Similarly, no and even not a faint Hoechst signal of the prospective CD45+ immune cell can be seen in the fourth row. Thus, it is difficult for me to conclude whether these Hoechst signals were really attributed to immune cell DNA in renal cancer cells. However, it cannot be ruled out that the weak Hoechst signal was lost due to data compression. Data presented in the third row also support the assumption that an intact CD45+ immune cells that was freshly engulfed/ phagocytosed by a renal cancer cell. However, the shape of the Hoechst and CD45 staining in Fig. 5B does not really fit with the appropriate data in Fig. 5A. It looks as if the picture has been rotated 90° to the left. For instance, a faint Hoechst and CD45 stain can be seen in the upper right corner, which, however, cannot be seen in the appropriate image of Fig. 5A. This must be checked and corrected.”

Thank you for your feedback regarding Figure 5. We have re-imaged the original samples so that we can show a more detailed “3D” depiction of these subcellular bodies. Figure 5 now shows max intensity projections of several Z-stacked images. Additionally, we show the orthogonal view of the planes containing trogosomes. These views definitively show that these structures are located within the RCC cells and that they contain DNA through Hoechst labeling.

“7) Transmission images should be provided for all immunofluorescence data.”

Unfortunately, we do not have transmitted light images to accompany the immunofluorescence data presented in this manuscript. However, we believe that, with the additional Z-stack images, the immunofluorescent images are sufficient to support the claims made in this manuscript.

Thank you again to our editor and reviewers. We are incredibly grateful for each of your time and expertise in reviewing our manuscript. We were very pleased with the feedback we received, and we believe that it has greatly strengthened our manuscript. We are hopefully that you will find the revised version satisfactory.

Sincerely,

Alfred Bothwell, PhD

Dr. Alfred L.M. Bothwell, Ph.D.

Professor, Department of Pathology, Microbiology and Immunology

---

## [Decision Letter · Decision Letter 1]

6 May 2025

Renal cancer cells acquire immune surface protein through trogocytosis and horizontal gene transfer

PONE-D-24-53647R1

Dear Dr. Bothwell,

We’re pleased to inform you that your manuscript has been judged scientifically suitable for publication and will be formally accepted for publication once it meets all outstanding technical requirements.

Kind regards,

Kenji Fujiwara, MD, PhD, FACS

Academic Editor

PLOS ONE

Additional Editor Comments (optional):

Dear Dr. Alfred Lester Meador Bothwell.

Thank you for resubmitting your manuscript. One reviewer and I agreed to the acceptance. I reviewed the manuscript instead of 2nd reviewer although the one did not take the re-review. I think the authors responded appropriately to all questions from the reviewer

Yours sincerely,

Kenji Fujiwara

Academic editor

Reviewers' comments:

Reviewer's Responses to Questions

**Comments to the Author**

1. If the authors have adequately addressed your comments raised in a previous round of review and you feel that this manuscript is now acceptable for publication, you may indicate that here to bypass the “Comments to the Author” section, enter your conflict of interest statement in the “Confidential to Editor” section, and submit your "Accept" recommendation.

Reviewer #2: All comments have been addressed

2. Is the manuscript technically sound, and do the data support the conclusions?

Reviewer #2: Yes

3. Has the statistical analysis been performed appropriately and rigorously? 

Reviewer #2: Yes

4. Have the authors made all data underlying the findings in their manuscript fully available?

Reviewer #2: Yes

5. Is the manuscript presented in an intelligible fashion and written in standard English?

Reviewer #2: Yes

6. Review Comments to the Author

Reviewer #2: (No Response)

7. PLOS authors have the option to publish the peer review history of their article (what does this mean? ). If published, this will include your full peer review and any attached files.

**Do you want your identity to be public for this peer review?** For information about this choice, including consent withdrawal, please see our Privacy Policy .

Reviewer #2: No

---

## [Editor Report · Acceptance letter]

PONE-D-24-53647R1

PLOS ONE

Dear Dr. Bothwell,

I'm pleased to inform you that your manuscript has been deemed suitable for publication in PLOS ONE. Congratulations! Your manuscript is now being handed over to our production team.

Kind regards,

on behalf of

Dr. Kenji Fujiwara

Academic Editor

PLOS ONE